# What Are the Potential Roles of Nuclear Perlecan and Other Heparan Sulphate Proteoglycans in the Normal and Malignant Phenotype

**DOI:** 10.3390/ijms22094415

**Published:** 2021-04-23

**Authors:** Anthony J. Hayes, James Melrose

**Affiliations:** 1Bioimaging Research Hub, Cardiff School of Biosciences, Cardiff University, Cardiff CF10 3AX, Wales, UK; HayesAJ@cardiff.ac.uk; 2Graduate School of Biomedical Engineering, UNSW Sydney, Sydney, NSW 2052, Australia; 3Raymond Purves Bone and Joint Research Laboratories, Kolling Institute of Medical Research, Royal North Shore Hospital, St. Leonards, NSW 2065, Australia; 4The Faculty of Medicine and Health, The University of Sydney, St. Leonards, NSW 2065, Australia; 5Sydney Medical School, Northern, Sydney University, Royal North Shore Hospital, St. Leonards, NSW 2065, Australia

**Keywords:** nucleus, heparan sulphate, heparan sulphate proteoglycan, perlecan, syndecan, glypican, tensegrity

## Abstract

The recent discovery of nuclear and perinuclear perlecan in annulus fibrosus and nucleus pulposus cells and its known matrix stabilizing properties in tissues introduces the possibility that perlecan may also have intracellular stabilizing or regulatory roles through interactions with nuclear envelope or cytoskeletal proteins or roles in nucleosomal-chromatin organization that may regulate transcriptional factors and modulate gene expression. The nucleus is a mechano-sensor organelle, and sophisticated dynamic mechanoresponsive cytoskeletal and nuclear envelope components support and protect the nucleus, allowing it to perceive and respond to mechano-stimulation. This review speculates on the potential roles of perlecan in the nucleus based on what is already known about nuclear heparan sulphate proteoglycans. Perlecan is frequently found in the nuclei of tumour cells; however, its specific role in these diseased tissues is largely unknown. The aim of this review is to highlight probable roles for this intriguing interactive regulatory proteoglycan in the nucleus of normal and malignant cell types.

## 1. Introduction

Aims of this study.

This review highlights recent observations of perlecan in the nucleus and perinuclear regions of intervertebral disc (IVD) cells and poses questions regarding its putative functions as a nuclear heparan sulphate proteoglycan (HS-PG) (Figure 1).

### 1.1. Intracellular Perlecan—What Does It Do?

The observation of intracellular perlecan in IVD cells as discrete nuclear foci [1], and perinuclear vesicular deposits, possibly destined for transportation out of the cell, poses some intriguing questions.
(1)Is the structure of nuclear-associated perlecan similar to the perlecan that occurs in the pericellular and extracellular environment?(2)What are the specific functional roles of nuclear-associated and perinuclear perlecan?(3)What, if any, are perlecan’s interactive ligands within the nucleus and perinuclear regions of the cell?(4)How do (1)–(3) relate to disease processes such as those occurring in tumour development.

Experimental studies are warranted to answer these important questions; however, these have yet to be undertaken. Important clues as to perlecan’s potential roles in the intracellular environment can nevertheless be deduced to some extent from the substantial literature existing on nuclear HS-PGs in a number of cell types in health and disease [2,3,4,5,6,7,8,9]. Hopefully, the speculations we raise will stimulate researchers to answer some of these intriguing questions.

### 1.2. Biosynthesis of Perlecan

Like all proteoglycans (PGs), the biosynthesis of the perlecan core protein occurs in the rough endoplasmic reticulum (RER), and it is then transported to the Golgi apparatus where post-translational addition of glycosaminoglycan (GAG) chains occurs [10,11,12]. Perlecan is then exported out of the Golgi in vesicles to the extracellular environment or it is translocated to the nucleus or perinuclear regions. The perinuclear (Golgi) region of the chondrocyte is the major site of radio-sulphate incorporation [13,14] and of PG localization [11,15,16,17,18,19]. Addition of xylose to the PG core protein linkage region occurs in the Golgi apparatus [20], followed by step-wise addition of two galactose residues and a GlcA residue to complete the Xyl-Gal-Gal-GlcA linkage tetrasaccharide, which then acts as an acceptor molecule for the assembly of HS and CS chains through the sequential action of sulpho- and glycosyl transferases and addition of GlcA and GluNAc in HS or GlcA and GalNAc in CS [21,22]. Nuclear/perinuclear perlecan thus appears to represent a proportion of the endogenously produced perlecan, with the majority undergoing vesicular transport out of the cell to the PCM and ECM, where its roles have been extensively examined. Cell surface HS-PGs such as the syndecan family, however, can be translocated from the cell surface to the nucleus under certain circumstances. It is not known if cell surface perlecan can also be translocated to intracellular regions by a similar mechanism—its large size may argue against this possibility.

#### Heparan Sulphate Is a Highly Interactive Molecule

Heparan sulphate (HS) is assembled from 12 variably sulphated GlcNAc-GlcA disaccharides [23] that also undergo acetylation/deacetylation, and epimerization and structural inversion of *D*-GlcA to *L*-IdoA in variable regions along the HS GAG chain and *O*-2 sulphation of IdoA. *O*-3 sulphation also occurs in HS; this is relatively rare, but has been identified as a key interactive component of some HS sequences that regulate proteins such as antithrombin. HS chains contain areas that are highly modified and separated by regions of low modification. The diverse sulphated HS disaccharide presentations are complex and, along with acetylation/deacetylation, provide a high level of structural complexity and charge heterogeneity in HS, and consequently it has a biodiverse range of interactive ligands [24] (Table 1 and Table 2). HS is the most heterogeneous GAG. Listings in the HS interactome demonstrate that HS binds >400 bioactive proteins [25]. A murine study of acute pancreatic disease, however, recently identified 786 HS-binding proteins; thus, the HS-interactome needs to be updated [26,27,28]. Such HS binding proteins include growth factors, cytokines, morphogens, ECM proteins, cell adhesion molecules, proteases, and protease inhibitors [29]. A bioinformatics analysis of mammalian proteins expressing a heparin/HS-binding motif associated with the immune system identified 235 candidate proteins, with the majority of these being intracellular proteins [30,31]. Physical interactions of neurexins with leukocyte common-antigen-related receptor tyrosine phosphatases act as synaptic organiser proteins that are regulated by HS-PGs in many neural cell types, contributing to synaptic plasticity and cognitive learning [32,33].

The Kyoto Encyclopedia of Genes and Genomes (KEGG) is a collection of databases covering all aspects of genomes, biological pathways, diseases, drugs and chemical substances. KEGG is used in bioinformatics and data analysis in the areas of genomics, metagenomics, metabolomics, and omics studies dealing with molecular modelling, and simulation systems biology and translational research in drug design and developmental applications. The KEGG PATHWAY database core wiring diagram encompasses a collection of pathway maps that integrate data from genes, proteins, RNAs, chemical compounds, glycans and chemical reactions, and can include inputs from disease-associated genes and drug target data held in other arms of the KEGG database collection. The KEGG PATHWAY MAP can include data from metabolism, gene information processing covering transcription, translation, replication, and repair, environmental processing information (membrane transport, signal transduction), cellular processes (cell growth, death and membrane functions), organ systems (immune, endocrine, nervous systems), human diseases and drug development. The Gene Ontology (GO) arm of the KEGG database is a major bioinformatics initiative designed to represent all gene and gene product data across all species [25,35].

## 2. Perlecan’s Roles in Vascularised, Tensional and Weight Bearing Shear-Loaded Tissues

Perlecan is a modular, multifunctional HS-proteoglycan (HS-PG), with roles in ECM stabilisation and organisation, cellular proliferation and differentiation, which sequesters a number of growth factors including the FGF family, PDGF, VEGF, BMP-2 and BMP-4 and presents these to and activates their cognate receptors to promote tissue expansion and ECM remodelling processes in tissue repair and in development [59]. Perlecan is active in vascular and poorly vascularised tensional and weight-bearing cartilaginous tissues and during endochondral ossification resulting in expansion of the axial and appendicular skeleton [60]. Perlecan has been proposed to be a cell-signalling hub co-ordinating the action of a number of growth factors in tissue development and morphogenisis. Perlecan attached to the lumenal surfaces of blood vessels acts as a flow sensor signalling to the endothelial and vascular smooth muscle cells (SMCs) to regulate vasodilation, blood pressure and vasculogenesis [61]. Perlecan domain II binds LDLs and aids in the clearance of lipids from the circulation. Binding of LDL to SMC perlecan may also lead to membrane depolarization, vasoconstriction and a lowering of cGMP levels with attendant effects on SMC cell signalling.

While perlecan promotes endothelial cell proliferation, it has an opposing effect on SMCs [62,63,64,65,66]. From Table 2A,B, a number of GO terms are associated with HS, displaying an increased occurrence of HS in the search category, the term HS is enriched in the following biological process categories: locomotory behaviour (entry 6), cell migration (entry 9), cell motility (entry 12), regulation of cell proliferation (entry 13), Regulation of response to external stimulus (entry 15), cell–cell signalling (entry 19). In Table 2B, KEGG pathways enriched in the HS interactome include cytokine–cytokine receptor interaction (entry 22), focal adhesion (entry 24), pathways in cancer (entry 25), regulation of the actin cytoskeleton (entry 29) and MAPK signalling pathway (entry 33).

Perlecan also acts as a flow sensor in the lacuno-canalicular space, acting as an osteocyte mechanosensor that detects external loading through solute fluxes in the lacuno-canalicular space [67,68]. Osteocytes are bone mechanosensors with roles in the regulation of bone homeostasis through Ca^2+^ transit through calcium channels, and interactions with G-protein coupled mechanoreceptors [69]. Perlecan also has cytoprotective roles in the type VI collagen matrices that make up the chondron surrounding chondrocytes, intervertebral disc and meniscal cells through its ability to modulate the biomechanical loading experienced by these cell types in their weight bearing environments [70]. The activity that perlecan displays is dependent on the tissue context and the form of perlecan present in specific tissue niches and cellular environments. Post-translational modifications to perlecans HS chains or proteolytic degradation of its core protein can both significantly impact on perlecan’s functional properties in situ. Perlecan also contains a number of matricryptic modules which when released by proteases can also impact on tissue repair, angiogenesis [45,71] and inhibition of angiogenesis depending on the matricryptic module that is released [72]. These perlecan fragments influence cell adhesion, invasion, and angiogenesis; however, it is not known to what extent these perlecan fragments are active intracellularly, but this remains a distinct possibility. In the tumour environment, protease-mediated changes in perlecan have been proposed to represent a molecular switch whereby an environment hostile to tumour development is transformed to one which promotes tumourigenesis [73].

Perlecan domain V (LG1LG2LG3) promotes tissue repair and angiogenesis and can act as a proteoglycan in its own right, while the LG1LG2 and LG3 fragments of perlecan domain V inhibit endothelial, α2β1 integrin and VEGFA interactions required for blood vessel formation, and these fragments are of interest as anti-tumour therapeutic agents [74]. It is paradoxical that intact perlecan promotes the development of a vascular supply that supports the proliferation of tumour cells and the development of a number of cancers while bioactive perlecan fragments inhibit tumour development by targeting its vascular supply [75,76,77].

## 3. Mechanosensory Processes in Tensional and Weight Bearing Tissues Effect the Nucleus and Impact on Gene Expression

The tensegrity theory, originally proposed by Ingber and colleagues in the 1990s [78,79], suggested that mechanical forces in tissues experiencing weight-bearing, tension and shear stresses could be transmitted from the ECM through to the PCM, and then to the cell cytoskeleton [80], and could be sensed by the nucleus with resultant effects on gene expression, and morphological development through mechanotransductive effects on cell signalling. Such intracellular mechanical forces alter the structure of the cytoskeleton [81,82,83,84] and regulate cell growth, migration, and tissue patterning during morphogenesis [84,85,86]. Since these earlier studies, the nucleus has been proposed as a mechanosensor [87,88]. Mechanical transductive effects [89] modulate cytoskeletal structures such as microtubules, microfilaments and the actin cytoskeleton [90,91,92,93,94,95]. Such cytoskeletal remodelling has been proposed to contribute to how early cellular life evolved [96], providing a means of altering cell shape and a means of modulating focal adhesion complexes required to allow cells to adhere to ECM components and undergo cell migration essential for tissue development [97,98]. Perlecan also has modulatory biomechanical roles in the ECM and PCM of tissues subjected to weight bearing, tension and shear forces [99]. Alterations in the ionic balance of lipid bilayers and the evolution of membrane polarization was also a means whereby the control of cellular behaviour evolved [100]. Cell membrane-associated GAGs and PGs have important roles to play in determining the resting potential of cells and the membrane polarization that initiates physiological cellular processes.

## 4. Membrane Polarization, Evolution of Membrane Energetics and Roles for GAGs in Motive Proton Gradients

Cellular activity powered by proton gradients has ancient roots [101] linked to the origins of life [101,102]. The harnessing of electrochemical ion gradients that traverse membranes to drive metabolic processes are as universally conserved as the genetic code [103]. Proton gradients are used by mitochondria to drive the synthesis of ATP, a molecule fundamental to energy production in higher animals. GAGs evolved through natural selection processes over a 500 million-year period of vertebrate and invertebrate evolution to develop proton detection and electroconductive capability [104] generating signals through interactions with protons to form the machinery of cell-signalling [105,106]. Proton gradients evolved as essential regulatory components of lipid bilayers and the origin of membrane bioenergetics [107] essential to the functional properties of cell membranes in higher animals [108,109]. Membranes become polarized through attainment of a positive charge on the outside of the cell and a negative charge in the inner aspect of the lipid bilayer. This is achieved by controlling the movement of ions passively and by gated ion channels to create action potentials central to cell–cell communication. Co-ordination of the opening and closing of voltage-gated ion channels controls the influx and efflux of ions (Ca^2+^, H^+^, Na^+^, K*+*), regulating membrane polarization. These processes are highly evolved in neurons which harness these processes in signal transductive neural networks, but all cells utilize similar processes in cell signalling to some degree during proliferation, migration and cell adhesion.

### Phosphatases/Kinases and HATs/HDACs in Cell Signaling and Gene Regulation

Protein kinases and phosphatases catalyse the formation or hydrolysis of phosphate groups, the transfer of phosphate groups to proteins or the hydrolysis of ATP to produce energy, and are fundamental to life processes. Both of these enzyme classes act as phosphotransferases, but have opposing modes of action. Kinase genes constitute only 2% of the human genome but they phosphorylate >30% of all human cellular proteins [110,111]. Phosphorylation is a ubiquitous regulatory mechanism mediating signal transduction in development, transcription, immune responses and in metabolic energy generation, apoptosis, and cell differentiation. Several classes of protein kinases act specifically on serine/threonine or tyrosine residues in proteins during cell signalling. Phosphorylation of biomolecules by protein kinases is a fundamental aspect of cell signalling in higher animals. Comprehensive cataloguing of the protein kinases of the human genome (the protein kinase kinome) was completed in 2002, with the identification of 518 protein kinase genes [112]. The human and murine kinomes have undergone proteomic analysis and these data are now listed on the UniProtKB/Swiss-Prot protein database [113]. Histone acetyl transferase (HAT)/histone deacetylase (HDAC) enzyme systems are also important regulatory enzymes that operate at the gene level and, as with phosphatases/kinases, are tightly regulated, since their aberrant regulation is implicated in many disease processes. GAGs inhibit HATs, resulting in the compaction of the chromatin structure and a reduction in transcription factor access to DNA. HS is the most active GAG inhibitor of the HATs [114]. Nuclear perlecan located in precise foci may act in conjunction with the HATs/HDACs to regulate chromatin structure, transcription factor activity and gene regulation. The importance of HS in these processes is exemplified by nuclear heparanase activity which mediates loss of nuclear Sdc-1, enhancing HAT activity and promoting the expression of genes that drive an aggressive tumourigenic phenotype [115].

## 5. Inhibition of Histone Acetyl Transferases and the Use of HDACs as Therapeutic Agents

GAGs are potent inhibitors of p300 and pCAF HAT activities in vitro, with heparin and HS-PGs being the most potent inhibitors [114]. Histone acetylation involves the addition of an acetyl moiety to the terminal ϵ-amino lysine residues of the tails of core histones, neutralizing the histones’ positive charge, lessening electrostatic interaction between DNA and histones, and forming a more open chromatin structure accessible to transcription factors. HDACs have an opposite mode of action to HATs resulting in the compaction of chromatin, reducing access to DNA for transcription factors. HDACs have roles in cartilage homeostasis [116] but may also be tumourigenic, leading to the development of therapeutic HDAC inhibitors [117,118]. Modification of proteins by HATs/HDACs is important in the regulation of gene expression, and dysregulation of this process is linked to malignant transformation and certain diseases [119]. The recent discovery of nuclear perlecan in IVD cells along with HDACs suggests that these act in concert to regulate chromatin compaction, gene regulation and positively contribute to tissue homeostasis [1].

### Histone acetylation/Deacetylation of Chromatin Effects Nucleosomal Structure and Impacts on Chondrocyte Regulation

Control of acetylation/deacetylation by HATs and HDACs affects important physiological and pathological cellular processes [120,121]. Acetylation of histones regulates gene expression through influencing chromatin architecture, regulating gene expression by opening or closing the chromatin structure to allow transcriptional access in cell cycle progression, proliferation and differentiation. Mounting evidence shows that nuclear HS and HS-PGs have regulatory functions impacting on the cell cycle, proliferation and transcription through their ability to influence chromatin structure in many cell types [8]. Nuclear HS-PGs also inhibit DNA topoisomerase I activity, regulating DNA removal from supercoils during transcription and DNA replication, re-annealing of DNA strands following strand breakage during re-combination and chromosomal condensation, and in the disentanglement of intertwined DNA strands during mitosis [122,123]. HS charge-mediated interactions regulate chromatin structure, nucleosomal function and gene expression [114]. It has yet to be ascertained what specific roles perlecan plays in such processes.

HDAC4 represses the attainment of chondrocyte hypertrophy, preventing endochondral bone development by inhibiting the function of myocyte-specific enhancer factor 2C (MEF2C) and runt-related transcription factor 2 (RUNX2) [124]. HDAC4 is the most extensively characterized member of the HDAC class IIa deacetylase enzyme family, which regulates signalling networks, affecting cartilage maturation. HDAC4 promotes chromatin condensation, reducing access of transcriptional factors to DNA and repressively regulating chondrocyte maturation and hypertrophy, and the deletion of HDAC4 results in premature calcification of cartilage [125]. HDAC4 null mice have squat runt-like frames—shortened growth plate hypertrophic zones, enhanced vascular invasion and cartilage mineralization. *MMP-13*, *Runx2*, *OPG*, *CD34* and *Wnt5a* are also down-regulated and type X collagen elevated in HDCA4 null mice. Adenoviral-mediated transduction of HDAC4 ameliorates disease progression in a rat OA model [126], lowering the MEF2C and RUNX2 activity that contributes to cartilage degeneration [127,128]. Increased HDAC2 activity in OA patients enhances cartilage degradation and represses cartilage-specific gene expression [129]. MicroRNAs that inhibit HDAC2 and HDAC3 [129,130,131] also downregulate ADAMTS-4 and 5 expression in IL-1β-mediated catabolism of human articular cartilage [129]. Based on the above information, future studies on the role of nuclear perlecan which could act in concert with the HDACs are warranted. Evidence from chondrocyte and cancer studies [132] indicate that this area of cell regulation may represent a new route of therapeutic intervention [119,133,134,135].

## 6. Cytoskeleton Mediated Spatial Re-Organisation of Cellular Components in Pre-Motile Cells

For cell migration to proceed, cellular polarization occurs in the leading and trailing edges of the cell, the nucleus is also re-positioned towards the back of the cell and the Golgi apparatus and centrosomes are moved toward the leading edge of the cell [136]. An asymmetrical Ca^2+^ gradient is also created from the back to front of the cell to regulate assembly of focal adhesions and promote migration. The polarized distribution of Sdc4 during cellular migration has clear roles in the migratory process, since Sdc4 KO cells exhibit decreased movement.

The establishment of cell polarity in migrating fibroblasts is essential for cell migration and precise spatiotemporal coordination of signalling pathways producing an asymmetrical profile with the nucleus re-located to the rear of the cell. Microtubule-mediated central re-positioning of the nucleus and the migrating cell edge establishes front-rear polarity and directional migration. The nuclear axis also requires alignment with the axis of cell migration for motility to occur. This re-orientation of the nucleus occurs through physical interconnections between the nucleus and cytoskeleton, termed a linker of the nucleoskeleton–cytoskeleton complex (LINC) [136], and is mediated by activation of GTPase Rho, integrin, focal adhesion kinase (FAK), Src, and p190RhoGAP signalling pathways. Spatial induction of integrin signalling at the leading edge of the cell and FAK and p190RhoGAP activation drives cell migration and is influenced by intracellular HS-PGs (Figure 2). As already noted, Sdc-4 has roles in these processes; however, perlecan also has multifunctional interactive properties (Table 1 and Table 2), suggesting that it may also participate in such processes.

In polarized motile fibroblasts, stress fibres have specific 3D orientations. Ventral stress fibres attach to focal adhesions at both ends on the basal side of the cell, while dorsal stress fibres, transverse actin arcs, and perinuclear actin fibres attach to the cell migration front [137]. Perinuclear actin fibres induce rotational movement of the nucleus, aligning it with the direction of migration. This network of dorsal fibres, transverse arcs, and perinuclear fibres transfers mechanical signals between the focal adhesions and nuclear envelope to regulate nuclear reorientation in polarizing cells. HS-PGs contribute to this process, with Sdc4 displaying specific localisations in migratory polarized cells linked to these re-positioning processes mediated by cytoskeletal components.

## 7. Structural Organisation of the Nucleus

### 7.1. Nuclear HS-PGs

Nuclear HS-PGs have previously been demonstrated in a number of cell types [138,139,140,141] and correlated with cell proliferation [138]. Nuclear glypican (Gpc) has been found in neurons and glioma cells, syndecan-4 (Sdc-4) in cardiomyocytes [142], Sdc-1 in fibrosarcoma cells [143], and Sdc-2 in the nucleus of tumour cells in osteochondromas [144]. The function of nuclear Sdc-1 is unclear; however, it may inhibit histone acetyltransferase, leading to compaction of chromatin, decreasing DNA accessibility to transcription factors, affecting cell cycle control, decreasing proliferation and reducing protein transcription and transport to the nucleus. HS-PGs display shuttling properties for protein kinase C-dependent nuclear translocation of FGF-2 in corneal trauma [139]. Nuclear HS-PGs inhibit DNA topoisomerase I, affecting the removal of DNA supercoils during transcription and DNA replication, regulating the re-annealing of DNA strands following strand breakage, re-combination and chromosomal condensation; and disentanglement of intertwined DNA strands during mitosis [122,123]. HS has previously been shown to modify the chromatin structure in the nucleosome and have potential effects on gene expression mediated by transcription factors.

### 7.2. Nuclear Heparanase Regulates HSPG Structure and Modulates Gene Activity

Degradation of nuclear Sdc-1 by heparanase enhances histone acetyltransferase activity and promotes expression of genes that aggressively drive the attainment of a tumour phenotype. Prolyl-4-hydroxylase controls the transactivation of NF-κB/p65 [145] and enhances the catabolic effects of inflammatory cytokines on cells in the NP. HIF-1 and HIF-2 control the expression of the prolyl hydroxylases (PHDs) in NP cells [146].

### 7.3. HS-Proteoglycan Mediated Interactions with the Cytoskeleton

Sdcs have roles in many disease processes, including the spread of tumours, inflammatory disorders, and arthritis [7]. The cytoplasmic domains of the cell surface Sdcs have interactive properties with the actin cytoskeleton and can bind and activate cell signalling molecules, thus the Sdcs, besides acting as cell surface PGs, are also receptors that initiate cell signalling [147]. Members of the Sdc family are also translocated to the nucleus [4,7,9]. Sdc-4 binds to the cytoskeleton through α-actinin and regulates signalling through protein kinase C and the RhoA and Rho kinases [148,149]. Sdc-4 may have tension-sensing roles, detecting changes in the cytoskeleton in wound contraction, interactions between tumours and surrounding stromal tissue, tissue fibrosis, cellular adhesion and pre-motile changes in cells [148,150]. Sdc-4 participates in cytoplasmic interactions that orchestrate adhesion and growth factor receptor signalling [151]. Sdc-4 modulates cell polarity and cell migration by influencing the positioning of the centrosome in pre-motile cells [136] and the reorientation of cell nuclei prior to establishment of front-rear polarity in migratory cells [152]. Sdc-4′s interactions with dorsal stress fibres, transverse actin arcs, and perinuclear actin fibres form an interconnected network that induces nuclear movement in polarizing pre-motile fibroblasts [137]. Dynamin II interacts with Sdc-4 to regulate focal adhesion and stress fibre formation in migratory cells [153]. The cytoplasmic domain of the cell surface Sdcs consists of two conserved regions (C1, C2) and a variable region (V). Contractin interacts with the C1 region, regulating formation of the cytoskeleton, protein secretion, GTPase-mediated cell signalling and cellular migration [154]. *Src*, proto-oncogene tyrosine protein kinase, a non-receptor protein tyrosine kinase, also interacts with the C1 domain and promotes cell signalling pathways that control gene transcription, immune responses, cell adhesion, cell cycle progression, apoptosis, migration, and cellular transformation [155]. Sdcs also have roles in Wnt signalling cascades [156]. Syndesmos interacts with the variable region of the cytoplasmic domain of Sdc-4, altering cytoskeletal structure and mediating cell spread [157]. The members of the α-actinin protein family interact with the variable region of the Sdcs, affecting actin cross-linking and promoting cytokinesis, cell adhesion and cell migration [158]. Synectin interacts with the C2 domain of Sdc-4 and inhibits cell migration [159]. Syntenin, a multifunctional Sdc binding adaptor protein, interacts with the C2 region, regulating cell signalling including the Wnt pathway [160,161]. The peripheral plasma membrane protein CASK, a multifunctional scaffolding protein, interacts with the C2 region, regulating neural synaptic activity. Synbindin is a cytoskeletal neuroregulatory C2 binding protein in Sdc-2 [162]. It is not known if the C1, C2 and V regions of cytoplasmic Sdcs interact directly with genes or with adaptor proteins or in a similar manner to cell membrane Sdcs (Figure 3).

### 7.4. Nuclear Protein Interactions with HS-PGs Effects Cytoskeletal Organization

When shed from the cell surface, the ectodomains of the Sdcs are endocytosed and they interact with cytoskeletal components [148,149]. Some CD44 isoforms also interact with cytoskeletal components [165]. Sdc1 regulates signalling pathways that control proliferation and migration of malignant mesothelioma cells and other cancers, translocating to the nucleus through a tubulin-dependent transport mechanism. Co-immunoprecipitation experiments with Sdc1 in mesothelioma cells have identified a large number of interactive proteins [5]. Proteomic analysis focussing on Sdc1 interactive nuclear proteins revealed pathways regulating cell proliferation, RNA synthesis, splicing and transport. A proteomic map of Sdc1-nuclear interactive proteins identified a previously unknown role for Sdc1 in RNA biogenesis. Transcriptome and proteomic analysis of fibrosarcoma cells in which nuclear translocation of Sdc1 occurs has revealed the activation of the TGF-β pathway and altered expression of early growth response 1 (EGR-1), never-in-mitosis gene a-related kinase 11 (NEK11), and dedicator of cytokinesis 8 (DOCK8) genes that are coupled with growth and cell-cycle regulation. Nuclear translocation of Sdc1 also alters the activity of the transcription factors E2F, NFκβ, and OCT-1. The transcripts and proteins affected by Sdc1 translocation are thus dominated by effects on protein phosphorylation and post-translational events due to alterations in intracellular signalling. Addition of exogenous HS or CS to arginine-rich peptide-DNA polyplexes leads to an increase in their gene delivery efficiency, improved intracellular routing and nuclear accumulation. This may explain the occurrence of HS-PGs in the nucleus of multiple cell types [166]. Shed Sdc1 is present in high levels in many tumour cell types where it shuttles growth factors to the nucleus by altering histone acetylation in host cells [9]. Heparanase regulates Sdc-1 levels in the nucleus [2], and reduction in nuclear Sdc-1 levels by heparanase enhances histone acetyltransferase activity, inducing aggressive genes that promote tumourigenesis. Dynamin II interacts with Sdc-4, a regulator of focal adhesion and stress fibre formation [153]. Intracellular ligands also promote nuclear translocation of FGFR1 [167]. Nucleolin is a nuclear target for the internalisation of Gpc-1 [168].

The nuclear envelope, a double membrane structure, encloses the nucleus and is punctuated by holes known as nuclear pore complexes (NPCs) [169]. The NPC is a massive (110 MDa in humans) octameric structure consisting of more than 100 proteins called nucleoporins [170], which function as transport routes for ions and macromolecules into and out of the nucleus and have roles in mitotic events [171,172] and cellular regulation [173,174,175,176]. The identification of perlecan, a PG with a 467 kDa core protein, in distinct foci in the nucleus of IVD cells suggests that the upper limit for entry of macromolecules into the nucleus through NPCs may be much higher than previously reported [170,177]. The NPC contains a large central channel, ~7 nm in width and 50 nm in length, which allows entry into and out of the nucleoplasm. NPCs display morphological and functional plasticity, adopting various shapes depending on their contextual environment [177].

## 8. Control of Chromatin Structure and Gene Regulation by HS-PGs

Chromatin is the principal component of the cell nucleus and is a complex arrangement of DNA and histones (H1-H4) that organise DNA into structural units called nucleosomes. Two H3 and two H4 proteins initially form a tetrameric structure, to which are attached two H2A/H2B dimers to form the core histone structure, while ~150 base pairs of DNA wrap around this to make a nucleosome core particle along with linker histones such as H1. Histones condense the chromatin structure, resulting in nuclear DNA, becoming wrapped around histones in the nucleus as chromosomes. Histones are basic proteins that interact with negatively charged DNA [169], some eight histone octameric complexes of doublet H2A, H2B, H3, and H4 appear as spools around which the thread-like DNA wraps itself to form “ beads on a string” structures [178]. The nucleosomes are wrapped into 30 nm spirals called solenoids containing additional H1 linker histones which maintain chromatin and the chromosome structure. Exposed N- and C-terminal histone tails in these structures regulate chromatin organization [178,179]. Nucleosome assembly involves interactions between positively charged histone and negatively charged DNA [180]. Neutralization of positive charges or the introduction of negative charges on histone H3/H4 tails, by acetylation or phosphorylation or attachment of HS/HS-PGs, weakens the histone–DNA interaction, leading to relaxation of the chromatin structure and increased accessibility to transcriptional factors [169]. Interaction of RNA with histone tails also promotes an open chromatin structure [181]. Control of the chromatin structure is normally highly regulated, and acetylation/deacetylation are tightly controlled processes. Aberrant control of these processes can deleteriously affect gene expression and lead to certain diseases including tumour development [182,183,184], and thus it is important that the correct chromatin structure is maintained [185]. The presence of nuclear HS-PGs has been demonstrated in many studies over the last five decades [138,139,140,141], and in many cases correlated with cell proliferation, suggesting that they interact with FGFs and FGFRs to initiate cell signalling [138]. Nuclear HS-PGs inhibit DNA topoisomerase I activity, which has important roles to play in the removal of DNA supercoils during transcription and DNA replication, in the re-annealing of DNA strands following strand breakage during re-combination and chromosomal condensation, and in the disentanglement of intertwined DNA strands during mitosis [122,123]. HS charge-mediated interactions represent a potential means whereby the chromatin structure, nucleosomal function and thus gene expression are regulated.

## 9. Nuclear FGF-1, FGF-2 and FGFRs

FGF-1 and FGF-2 are major perlecan ligands in the PCM/ECM where they are chondroprotective, mechanotransductive agents. Perlecan acts as a low affinity co-receptor for the FGFs and activates FGFRs, leading to signal transduction and cell signalling. Nuclear/perinuclear perlecan may display similar interactive properties with nuclear FGF-1, 2 and FGFR1, but studies are warranted to confirm this possibility. The nuclei of quiescent cells do not contain FGFR1; however, cells treated with FGF-2 display a dose- and time-dependent increase in nuclear FGFR1. Cell-surface FGFR-1 labelled with biotin is detected later in the nuclear fraction of FGF-2-treated cells, indicating that nuclear FGFR-1 is translocated from the cell surface [186]. The identification of perlecan as distinct foci in the nucleus, a mechanosensitive organelle, is an intriguing observation given the known matrix stabilising properties that perlecan has in tissues and its participation in biomechanical processes and mechanotransductive cell signalling events initiated outside the cell. However, the localisation of perlecan in the nucleus does not automatically demonstrate that it has roles in the stabilisation of nuclear architecture. The nucleus has a sophisticated structure containing an interactive nuclear envelope and cytoskeletal proteins with established dynamic nuclear mechano-supportive roles, and roles for perlecan in such processes may thus be redundant. Perlecan may have other functional roles in the nucleus unrelated to its roles in the extracellular environment. Four structurally related intracellular non-signalling FGFs have been identified that interact with voltage-gated ion channels to regulate intracellular sodium levels.

The FGF family has 23 members, and these interact with tyrosine kinase receptors FGFR1-4 to initiate cell signalling. Activated FGFRs phosphorylate specific tyrosine residues, mediating interactions with cytosolic adaptor proteins in the RAS-MAPK, PI3K-AKT, PLC, and STAT intracellular signalling pathways. A single FGF2 transcript can be translated into five FGF2 protein isoforms, an 18 kDa low molecular weight secreted isoform and four larger 32–34 kDa high molecular weight isoforms [187,188,189] that are not generally secreted but have intracellular properties including roles in ion channels and stem cell renewal [190]. Nuclear FGFR and polypeptide growth factor signalling regulates skeletal development and disease processes [191,192]. FGFRs can enter the nucleus during the cellular transition between proliferation and differentiation where they interact with chromatin remodelling proteins, altering the epigenetic state and transcriptional status of target genes. The FGFRs are known by several alternative names, illustrating their biodiverse biological areas of interaction (Table 3).

### Neuronal Cell-Repair Responses Initiated by FGF-2, FGFR-1 and HS-PGs

In the intact cerebral cortex, FGF-2 and FGFR1 mRNA and protein are constitutively expressed by astrocytes and neurones, respectively, and FGF-2 protein is localized exclusively to astrocyte nuclei. FGF-2 signals through FGFR-1 using HS-PGs as co-receptors. Examination of FGF-2, FGFR1, Sdc-2 and -3, Gpc-1 and -2, and perlecan in neurones and glia in and around adult rat cerebral wounds shows that FGF-2 mRNA is up-regulated only in astrocytes and FGFR1 mRNA expression increased in glia and neurones. FGF-2 may act as a paracrine autocrine neuron and glial factor. FGF-2 protein localizes to the cytoplasm and nuclei of injured neurones and glia with weak or no staining of HS-PGs in normal cerebral glial nuclei, and only a few immunopositive neurones. Differential co-localization of HS-PGs, trafficked intracellular FGF-2 and FGFR1 after injury indicates that FGF-2-FGFR1-HSPG complex formation regulates FGF-2 storage, nuclear trafficking and CNS cell-specific injury responses.

## 10. Investigations on Nuclear Interactomes Reveals the Roles of Nuclear HS-PGs

Recent studies evaluating nuclear interactomes represent a potential new gene regulatory area for therapeutic targeting of cells in disease processes. Proteomic analysis of the nuclear Sdc-1 interactome of mesothelioma cells identified proteins interacting with nuclear proteins and associations with pathways related to cell proliferation, RNA synthesis, splicing and cellular transport [196]. The cardiac SDC-4 interactome has also been evaluated and 21 novel and 29 previously described interaction partners have been identified, including mechanotransducer muscle LIM domain proteins which regulate myocyte differentiation and have roles in the conductive properties of cardiac tissues. Assessment of the RNA polymerase-I RNA interactome showed this to be highly enriched in nucleolar proteins associated with ribosome biogenesis and RNA binding activity dependent on RNA polymerase-I activity [142]. The RNA interactome is important in the regulation of chromatin-nucleosomal organisation. Nuclear HS-PGs have been proposed to have regulatory roles in nucleosomal modifications that regulate transcription factor activity and gene regulation. Many of these interactions involve the HS chains of the HS-PGs, which explains the gene regulatory effects reported for nuclear heparanase [197,198,199,200,201] and the association of HS-PG expression with tumour development.

## 11. Changes in Chromatin Structure in the Nucleus—Can Perlecan Stabilise Chromatin?

The nucleus of mammalian cells is surrounded by a nuclear envelope containing two membranes and nuclear pores, an aqueous channel for bidirectional transport of ions and macromolecules between the nucleus and the cytoplasm [170,177]. Chromatin’s structure is highly sensitive to its ionic microenvironment. In order for DNA to fit into the limited space of the cell nucleus and maintain its dynamic accessibility for transcription, replication, repair and recombination, it exists as a complex with nuclear histones, facilitating folding, compaction and DNA accessibility in the nuclear chromatin. Ionic environments influence the compaction of chromatin under physiological conditions. K^+^, Mg^2+^ and Na^+^ are the main cytoplasmic cations. In the presence of Mg^2+^, Na^+^ ions promote folding of beads-on-a-string nucleosomal arrays into 30 nm fibres, whereas K^+^ and Mg^2+^ abrogate this process, demonstrating the complexity of the regulation of dynamic chromatin compaction in vivo [202]. The GAG chains of HS-PGs such as perlecan carry counter-ions and thus may act as ion reservoirs in discrete locations in the nucleus, and thus may regulate the nuclear ionic microenvironment. Chromatin fibres stabilise nucleosomes under torsional stress [203]. HS-PGs counter tensional and shear stresses in connective tissues, and thus their localisations in precise regions of the nucleus may also safeguard chromatin from excessive supercoiling during DNA replication through interactions between basic chromatin and negatively charged GAG chains on perlecan [204].

## 12. Nuclear Mechanosensory Properties: Dynamic Interplay of Proteins in the Nucleoskeleton, Nuclear Envelope and Cytoskeleton Provide Mechanical Support to the Nucleus

The cell nucleus is surrounded and protected by the nucleoskeleton, a large network of physically interconnected structural nuclear proteins attached via cytoskeletal filaments to adhesion molecules as an integrated scaffold equipping the nucleus with an ability to cope with mechanical stress (Figure 2). This network also has mechanotransductional properties that allow the nucleus to perceive and respond to mechanical stress. Biophysical microenvironmental stimuli significantly impact on cell function and behaviour. The nucleus is surrounded by a nuclear envelope consisting of outer and inner nuclear membranes separated by a perinuclear space that has roles in the regulation of nucleoplasm–cytoplasm communication, and it also provides a scaffold for chromatin attachment, regulates chromatin dynamics during cell division and has roles in mechanotransduction [205,206]. The cell nucleus is the largest and stiffest organelle, and is connected to the cytoskeleton by the LINC complex [206]. The nucleus, nuclear envelope and associated cytoskeleton have roles in mechanotransduction pathways that regulate cellular activity [96]. Cell nuclei are mechanosensitive organelles that allow cells to perceive and respond to mechanical stimuli [207,208].

Nesprin1-4 (nuclear envelope spectrin repeat proteins) are a family of intracellular scaffolding proteins that are found primarily in the outer nuclear membrane [209,210,211]; they form part of the LINC complex connecting the nucleo- and cyto-skeleton through the nuclear envelope [206]. The LINC complex is a multi-protein structure involving interactions between emerin, lamin A/C, SUN1, SUN2, nesprin-1 and nesprin-2 in the nuclear envelope and with actin filaments and type B lamins. Since the LINC complex has roles in signalling pathways and gene regulation, aberrant assembly of the LINC complex or mutations in any of its many component proteins can lead to a number of human diseases [212]. The SUN-domain family of nuclear envelope proteins interact with KASH-domain partners to form SUN-domain bridges across the inner and outer nuclear membranes, physically connecting the nucleus to every major component of the cytoskeleton [213,214]. SUN-domain proteins are multifunctional mechanical adaptors and nuclear envelope receptors that have diverse roles in the positioning of the nucleus and its re-orientation during cell polarization prior to cell migration, the localization of centrosomes, telomere positioning and apoptosis. Barrier-to-autointegration factor (BAF) interacts with double-stranded DNA, chromatin, histones and nuclear envelope proteins, protects genomic DNA and has essential regulatory functions in cellular replication [215,216,217] (Figure 2).

The LINC complex has a broad range of functions besides the maintenance of nuclear architecture, nuclear orientation and migration during cell polarization, and also has modulatory effects on gene expression. SUN-domains also have roles in the positioning of the nucleus during the cell polarization preceding cell migration [213,214]. Kinesin Eg5 is a motor protein involved in the establishment of a bipolar spindle and is one of 45 kinesins encoded in the human genome. The mitotic spindle is a microtubule-based assembly that separates the chromosomes during cell division [218]. The lamin-A/C-LAP2α-BAF1 protein complex regulates mitotic spindle assembly and positioning [219]. Cytoskeletal actin has well-known structural roles controlling cell shape, but also participates in many processes in the nucleus [211]. Monomeric and polymeric actin occur in the nucleus, monomeric actin regulates gene expression through transcription factors, chromatin regulating complexes and RNA polymerases. Nuclear proteins, such as emerin, regulate actin polymerization in the nucleus, and polymeric actin has roles in nuclear organisation and maintenance of genomic integrity. Dynein is an anchorage protein that aids in the generation of traction forces transmitted through microtubules that facilitate the central positioning of centrosomes in cells [220]. Intermediate filaments resist tensile and compressive forces in cells [221]. They are crosslinked to each other and to actin filaments and microtubules by desmin, filamin C, plectin, and lamin (A/C) [222]. Formation of cytoplasmic and nuclear networks by intermediate filaments provide cells with mechanical strength, while abnormalities in the structure of these assemblies leads to cell fragility in a number of genetic diseases. Anchorage of intermediate filament networks to the nuclear envelope and the actin and tubulin cytoskeleton occurs through the cytolinker protein plectin. Emerin has diverse functions, including the regulation of gene expression, cell signalling, nuclear structure, chromatin tethering and chromatin architecture and mechanotransduction. Relatively little is known about many of the component proteins of the nucleoskeleton; however, emerin is one of a few nuclear membrane proteins where extensive knowledge on its biochemistry, interactive partners, functions, localizations, posttranslational regulation, roles in development and links to human disease is available [223,224]. Barrier-to-autointegration factor (BAF) interacts with double-stranded DNA, chromatin, histones and nuclear envelope proteins [215]. BAF appears to protect the genome and enables cell division [216]. LAP2 (thymopoetin), a lamin- and chromatin-binding nuclear protein, occurs as three alternatively spliced ubiquitously expressed cellular proteins of 75 kDa (alpha), 51 kDa (beta) and 39 kDa (gamma). LAP2 is the mouse homologue. LAP2 regulates nuclear architecture by binding lamin B1 and chromosomes. LAP2 interacts with BAF [217,225] and mediates membrane-chromatin attachment and lamina assembly [217,225]. As shown in Table 1 and Table 2 and as already discussed, perlecan interacts with a large number of ligands through specific core protein and HS interactions. Thus, while specific interactions for perlecan in the nuclear, perinuclear or cytoskeletal regions other than with chromatin have yet to be identified, it is highly probable that perlecan will interact with some of these components.

## 13. Nuclear HS-PGs and Tumour Development

The prominent localisation of HS-PGs in the nucleus of tumour cells promotes the development of many tumour cell types (Table 4). HS-PGs are also found in normal cell nuclei, indicating they also have cell regulatory roles in normal tissues. *Snail* signalling contributes to prostate cancer progression [226] and metastasis through nuclear translocation of the intracellular domain of Sdc-1 in prostate cancer cells. In order to metastasize, cancer cells often transition from an epithelial to a mesenchymal phenotype through an EMT transition, and modify the ECM, facilitating escape into the circulation and avoiding immune surveillance. Eventually, they migrate and adhere to distantly located tissue sites and form secondary tumours. Cell surface Sdc-1 on normal keratinocytes interacts with the LG4/LG5 domain of laminin-322, promoting cellular migration; similar interactions also operate in tumour cells [227]. Sdc2 promotes the invasive properties of lung adenocarcinoma cells—this effect is mediated by syntenin-1 [228]. Syntenin-1, mda-9 (melanoma differentiation-associated gene-9) is a multifunctional adaptor protein containing tandemly repeated PDZ domains that facilitate cytoskeletal attachment of Sdcs, regulating transmembrane receptor trafficking and tumour cell metastasis [229,230,231]. Syntenin-1 also promotes the metastasis of lung adenocarcinoma cells, colorectal tumour cells and glioma cells by promoting the generation of exosomes to propagate tumour cell development [232,233]. Nuclear translocation of Sdc-1 hampers the proliferation of fibrosarcoma cells by interfering with the cell cycle; however, proteolytic generation of a C-terminal fragment of Sdc-1 by ADAM 17 inhibits lung tumour cell migration but promotes lung epithelial tumour cell migration and metastasis [171] Shed Sdc-1 mediates tumour–host cell communication in myeloma cells by shuttling growth factors to the nucleus and by altering histone acetylation [9]. The importance of the HS chains of Sdc-1 in the prevention of tumour spread is exemplified by the heparanase-mediated loss of HS in Sdc-1, which enhances histone acetyltransferase activity and results in gene expression to drive an aggressive tumour cell phenotype. Colon cancer cells transfected with the syntenin-1 gene display increased migratory activity [228]; however, interaction of syntenin-1 with the cytoplasmic domain of Sdc-2 abrogates this migratory activity.

## 14. Perlecan’s Contributions to the Malignant Phenotype

### 14.1. Perlecan-FGF-FGFR Interactions

Abnormalities in nuclear morphology and architecture are commonly observed in aged and senescent cells and can lead to apoptotic changes and cell death [245]. In cancer, there is extensive re-organisation of nuclear structure and dynamic changes in genomic organisation which may lead to nuclear stabilisation and the ability of the cancer cell to avoid apoptotic changes and remain in a highly proliferative state. A total of 18 receptor tyrosine kinases (RTKs) are known to be trafficked from the cell surface to the nucleus, including FGFR1-3 [246].

Many of these nuclear RTKs, including FGFR1-3, are overexpressed in cancer. There are also many cases where HSPGs have been shown to stabilise nuclear structure or regulate gene expression [247]. Cell membrane FGFR has well-known cell-signalling properties that regulate cell proliferation and differentiation during development and homeostasis. However, nuclear FGFRs also have critical roles to play in cell proliferative processes [3]. Full-length FGFRs internalized by endocytosis or arising from de novo synthesis enter the nucleus by means of a β-importin-chaperone-dependent mechanism. Once inside the nucleus, FGFRs interact with chromatin remodelling proteins to alter the epigenetic state and transcriptional status of target genes. A large number of studies have demonstrated the translocation of FGFR1 from the cell surface to the nucleus [248]. FGFR-1 and its binding partner, cyclic amp-response element binding protein (CREB), act as a nuclear regulatory complex, interacting with and regulating numerous genes and diverse developmental signals [248].

Perlecan has well-known roles as a FGF co-receptor and also interacts with and activates FGFRs. The presence of perlecan in small nuclear foci is highly suggestive of a regulatory role in FGF–FGFR interactions that promote cell proliferation. This is in agreement with the properties of perlecan in the extracellular environment, where it promotes cellular proliferation and differentiation. There is a strong likelihood that nuclear perlecan also promotes FGF–FGFR interactive processes that induce cell proliferation, a hallmark of cancer, which may explain the association of perlecan with a large number of cancer types (Table 4).

### 14.2. Perlecan-HS Interactions with Histones

The sulphated GAGs of the CS and HS side chains of perlecan have interactive properties with the basic lysine and arginine residues of histones, which are extremely basic proteins and thus display a positive charge, explaining how electrostatic interactions with the negatively charged GAG chains of perlecan can mediate strong interactions with histones. In myoblasts, HSPGs modulate FGF-2 activity and regulate skeletal muscle differentiation [249]. In vitro binding assays have demonstrated that histone H1 binds specifically to perlecan. Immunofluorescence microscopy demonstrated an extracellular pool of histone H1 colocalized with perlecan in the ECM of myotube cultures and in regenerating skeletal muscle. Histone H1 in the ECM strongly stimulates myoblast proliferation via a HS-dependent mechanism. This is consistent with the cell proliferative roles attributed to perlecan in tumour development.

### 14.3. Perlecan-VEGF-2 Interactions Promote Vascularisation of Tumours

Perlecan also interacts with VEGF2 and promotes angiogenesis and the blood supply that is so critical to tumour development and metastatic spread [45].

### 14.4. Perlecan Contributes to Cancer Promoting Processes Proposed by Hanahan and Weinberg

Hanrahan and Weintrub [9] proposed that all cancer cells (i) require an ability to evade apoptotic changes, (ii) have a sufficient store of growth factors, (iii) display sustained angiogenesis, (iv) have limitless proliferative capacity, and (v) display tissue invasive metastatic properties that promote tumour development. Perlecan could potentially support all of these processes.

### 14.5. Perlecan and Potential Antagonism with Histone Transport to the Nucleus

Perlecan could potentially impact on gene expression through antagonising histone transport into the nucleus. The transport of histones from the cytoplasm to the nucleus of the cell, through the nuclear membrane, is a regulated cellular process required for the supply of new histones to the nucleus, essential for DNA replication and transcription [250]. Chromatin contains the core histone proteins H3, H4, H2A and H2B and the linker histone H1, which need to be transported from the cytoplasm into the nucleus [250]. Histones finetune the organisation and functional properties of chromatin [251]. Entry of histones into the nucleus does not occur by passive diffusion, despite the relatively small size of these proteins, but occurs by complexation of histones with a family of chaperone transport proteins that accompany the histones through the nuclear pore to the nucleus. These chaperones also ensure that inappropriate histone interactions with other proteins does not occur in the cytoplasm. It is interesting to ponder whether perlecan can potentially regulate or antagonise these histone interactions; if so, this would impact DNA replication and gene expression. In vitro binding assays have demonstrated that histone H1 binds specifically to perlecan, and thus there is the potential for perlecan to act as an intracellular competitive histone binding partner.

### 14.6. Electrostatic Interactions between Lysine and Arginine Histone Residues and Perlecan-HS

The interaction of HS with histones is a multi-point electrostatic interaction that stabilises chromatin structure. Depolymerisation of perlecan’s HS chains by nuclear heparanases or O-6 desulfation of HS by Sulf 1 and 2 would potentially regulate such interactions. Patients expressing high levels of heparanase display elevated expression of proteins involved in chromatin remodeling and several oncogenic factors compared to patients expressing low levels of heparanase [247]. This supports a role for heparanase in driving tumour progression. Heparanase promotes relaxation of chromatin structure and transcriptional activity. HSPGs also regulate chromatin structure but in an opposite manner, resulting in condensation and stabilisation of chromatin structure and inhibition of transcriptional activity [247]. Perlecan has been proposed as a molecular switch, acting initially to deter tumour development, but when it is modified by MMPs and heparanase it can be transformed into a tumour-tolerant matrix component [73]. Heparanase is preferentially expressed in neoplastic tissues and associated with histone modifications that contribute to tumour metastasis and angiogenesis [252].

## 15. HS Is Involved in Regulatory Processes in Cancer

### Histone Acetylation-Deacetylation, HS and Regulation of the Malignant Phenotype

Histones are the main structural proteins of chromatin in mammals. Histone acetylation/deacetylation is an epigenetic mechanism whereby regulation of gene expression is catalysed by histone acetyltransferases (HAT) and histone deacetylases (HDAC) [253]. The alterations in DNA structure that influence the activity of transcription factors, which induce or repress gene transcription. HATs catalyse acetylation and gene transcription and promote the transport of newly synthesized histones from the cytoplasm to the nucleus, whereas HDACs mainly silence gene expression and are promising anti-cancer agents.

Glycosaminoglycans inhibit HAT in vitro, with HS being the most potent inhibitor. The Sdc-1 ectodomain shed from the cell membrane by proteases can translocate to the nucleus of tumour cells and inhibit HAT and tumour development; however, this translocation process is blocked by HS [9]. Heparanase can be used to regulate Sdc-1 levels in the nucleus [2]; however, removal of HS from the nucleus by nuclear heparanase promotes an aggressive tumourigenic phenotype by enhancing HAT activity, which elevates the expression of tumour-promoting genes.

## 16. Potential of HS as a Therapeutic Target in Cancer

HS has emerging roles in oncogenesis of potential therapeutic value for human cancers as part of a complex HS signalling network [254]. Cell-surface HSPGs of the Sdc and Gpc families have key roles in the regulation of cell behaviour, cell signalling, and cell matrix interactions in normal and pathological tissues [255,256,257]. The soluble Sdc and Gpc ectodomains shed from cells by proteases act as circulating regulatory PGs for normal and tumour cells, where they become translocated into the nuclei of these cells [258]. Cell-surface Sdcs and Gpcs contain bound growth factors that migrate with the shed HSPG to be taken up by more distant cells where nuclear heparanase releases the bound growth factors, promoting cell proliferation and differentiation and the pathogenesis of certain disease processes in cancer and inflammation [259,260]. Key functional aspects of the HS component of these PGs are thus evident—6-O sulfation in particular is an important functional determinant of HS. Selective de-sulfation of HS by Sulf-2, a 6-O de-sulfatase enzyme, modulates ligand interactivity of HS and down-line cell signalling, contributing to improved survival rates in head and neck squamous cell carcinoma [261]. HS and HSPGs thus represent key therapeutic targets in the clinical treatment of cancer [262].

## 17. Conclusions

The observation of perlecan as a nuclear-associated/perinuclear component in normal cells of the IVD is an intriguing finding and opens up the possibility that perlecan may somehow modulate nucleosomal organization or effect transcriptional regulation and gene function. The presence of perlecan ligands (FGF-1, FGF-2, FGFR1) in nuclear locations suggests that they may also have roles in these regulatory processes. FGF-3, and FGF-11-14 are also found in the nucleus. A lot needs to be learnt of the potential roles of nuclear-associated perlecan, together with its interactive ligands, and thus further studies are warranted. Chip sequencing may be a useful approach to determine DNA binding sequences for perlecan. If perlecan directly regulates gene expression, this may represent a novel therapeutic pathway worthy of further exploration in the treatment of disc disorders and lower back pain. Nuclear perlecan is also frequently found in many tumour cells and roles in the aberrant regulation of gene expression in these cases may also be evoked. It is incongruous that perlecan should have been localised in nuclear and perinuclear locations in IVD cells that experience biomechanical stimulation. Connective tissue cells also contain a sophisticated dynamic mechanoresponsive cytoskeleton and nuclear envelope that protect the nucleus and allow it to perceive and respond to such environmental forces. While nuclear-associated perlecan has been observed, it cannot be presumed that it provides mechanical support in the nucleus in a similar manner to how it performs this role in the pericellular and extracellular environment. Further studies need to be undertaken to assess this possibility. It may well be that such roles for perlecan are redundant given the sophisticated dynamic cytoskeletal and nuclear membrane proteins and their roles in nuclear stabilization. Perlecan may therefore have totally unrelated properties which have yet to be discovered, in a similar manner to the non-signalling FGFRs, which have roles in the regulation of voltage-gated ion channels, rather than in growth factor signalling. HS and perlecan are, however, highly interactive with a wide repertoire of ligands, and the probability that HS binding proteins are also present in the nucleus and cytoskeleton is high. Experiments need to be conducted to ascertain this possibility. It is already known that perlecan HS chains interact with chromatin through electrostatic interactions, and this may explain its focal localization in the nucleus. Further studies therefore need to be undertaken to determine the significance of the nuclear localization of perlecan in normal and malignant cell types.

## Figures and Tables

**Figure 1 ijms-22-04415-f001:**
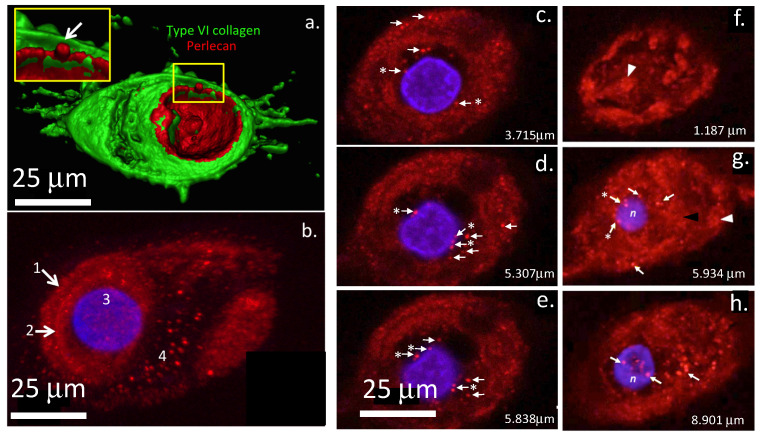
Immunolocalisation of perlecan and type VI collagen in a 3D surface rendered image of an ovine IVD chondron (**a**) and perlecan in a z-stacked image (**b**) and on 0.5 μm z-stacks at specific levels through additional cells (**c**–**h**). Fluorescently labelled inner AF cells were scanned on a Leica TCS SP2 AOBS laser scanning confocal microscope (Leica, Heidelberg, Germany) using 40× and 63× oil immersion objectives. Samples were scanned sequentially for DAPI (Ex max: 359; Em. Max: 461; blue nuclear fluorescence) and Alexa 594 (Ex max 594; Em max: 618; red perlecan fluorescence) or DAPI, Alexa 594 and Alexa 488 (Ex max 488 nm; Em max 510 nm; green type VI collagen fluorescence). Z-stacks of 8-bit ‘optical sections’ (512 × 512 pixels) were taken at 0.5 μm increments using Leica Confocal Software (Leica, Heidelberg, Germany). Confocal image datasets were also imported into Imaris for Cell Biologists software (Bitplane, Oxford Instruments) for 3D processing and fluorescent localisations. To conceptualise the fluorescent labelling patterns in 3D space, Z-stacks were modelled using maximum intensity and surface coding reconstruction algorithms [34]. A surface rendered image depicting type VI collagen, DAPI and perlecan are shown in a chondron (**a**). The inset depicts blebbing of the plasma membrane showing a vesicle containing perlecan being secreted from the cell. In (**b**), perlecan is immunolocalised in a z-stacked confocal image of a chondron. Perlecan is prominent in the PCM and type VI collagen capsule surrounding the cell and in vesicles throughout the chondron. In the inset image, perlecan is localized in the nucleus (n) in focal deposits in a non-stacked 0.5 μm confocal images. These deposits of perlecan are not evident in the stacked image in (**b**) due to overlying tissue. Perlecan is localized in (1) the type VI collagenous capsule, (2) pericellular matrix, (3) nucleus and (4) chondron surrounding the cell in (**b**). In single z-stacks (**c**–**h**), perlecan localization in the perinuclear region is labelled with an asterisked arrow, and in the nucleus and surrounding cell regions by an unlabelled arrow. The scale bar in (**e**) applies to all images in (**e**–**h**). Cell nuclei are stained with DAPI (blue) and perlecan with an anti-perlecan monoclonal domain IV antibody (MAb A7L6) and NovaRed substrate. Type VI collagen in (**a**) is labelled with a type VI collagen-FITC Ab. Images reproduced from [1] with permission.

**Figure 2 ijms-22-04415-f002:**
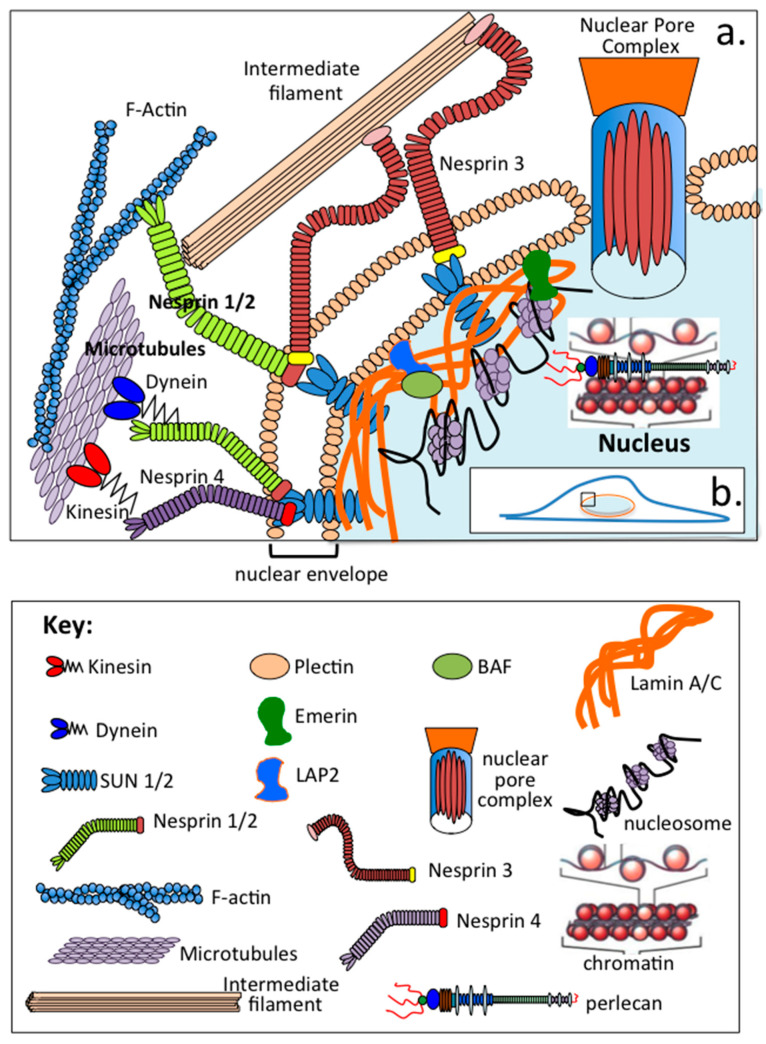
A simplified schematic depiction of structural proteins of the nuclear envelope and cytoskeleton that provide mechanical support to the nucleus. The nuclear envelope is a double membrane which contains proteins that attach the membranes to one another. These include the Sun 1/2 proteins, emerin, LAP2, and BAF, which also interact with LAM A/C nuclear proteins. The nuclear membranes are composed of plectin, nesprin ½ and nesprin ¾, which interact with the Sun proteins, and dynein and kinesin to attach the nuclear envelope to cytoskeletal components such as the actin skeleton, microtubules, and intermediate filaments. Nuclear pore complex (NPC) proteins are also present in the nuclear envelope and allow entry of proteins in and out of the nucleus. The NPCs are massive complex multi-protein complexes containing proteins arranged in an octameric arrangement around a central pore.

**Figure 3 ijms-22-04415-f003:**
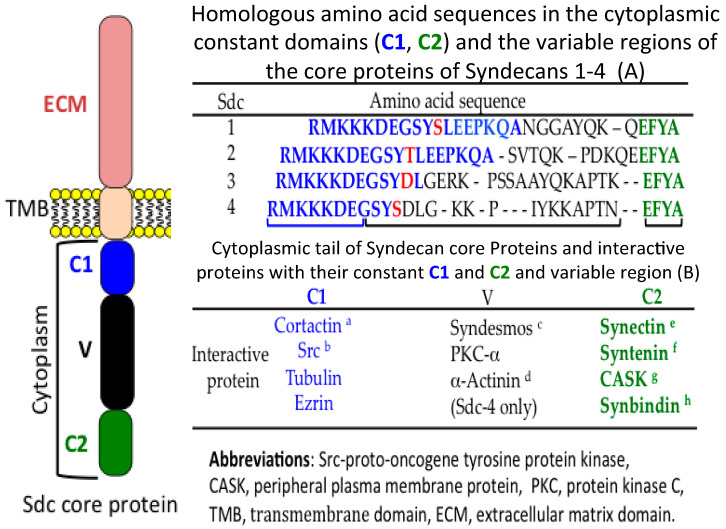
**The cytoplasmic domain of the syndecan family and its interactive proteins.** Amino acid sequences in the cytoplasmic domains of Sdc 1–4 contain conserved C1 and C2 regions and a variable region (V) interactive with a number of proteins. (a) Cortactin regulates protein secretion and activation of Rho-GTPases in vesicular trafficking, exocytosis, GTPase signalling, and transcription to regulate cell migration. Cortactin activates the actin-related protein 2/3 (Arp2/3) complex, and is regulated by post-translational modifications, including phosphorylation and acetylation [154]. (b) *Src*, proto-oncogene tyrosine protein kinase is a non-receptor protein tyrosine kinase that is activated following immune responses, integrin, adhesion, receptor protein tyrosine kinase, G protein-coupled and cytokine receptor mediated interactions in cell-signalling pathways that control gene transcription, immune responses, cell adhesion, cell cycle progression, apoptosis, migration, and cellular transformation [155]. (c) Syndesmos is a protein that interacts with the cytoplasmic domain of syndecan-4 and mediates cell spreading and actin cytoskeletal organization [157]. (d) α-Actinins are a major class of actin filament cross-linking proteins expressed in virtually all cells assisting in cytokinesis, cell adhesion and cell migration [158]. (e) Synectin is a PDZ protein involved in Sdc-4-dependent interactions and may have a role in the assembly of the syndecan-4 signaling complex and inhibition of cell migration [159]. (f) Syntenin is a multifunctional Sdc binding adaptor protein that regulates signalling pathways including Wnt signalling and cellular functions [160,161]. (g) The CASK gene is essential for normal neural development. CASK is a multidomain scaffolding protein that is highly expressed in the mammalian nervous system targeting neuronal synapses to regulate ion channel signalling [163]. (h) Synbindin is an atypical PDZ Sdc-2 binding protein [162,164].

**Table 1 ijms-22-04415-t001:** Examples of cell interactive HS binding proteins.

**Protein**	**Reference**
**Growth factors**	
EGF family	[36,37]
FGF family	[38,39,40,41,42,43,44]
VEGF	[45,46]
HGF	[47,48]
PDGF	[49]
TGF-β superfamily	[50,51]
**Cytokines/Chemokines/Morphogens**	
BMPs	[52]
CCL2	[53,54]
PF-4	[55]
HH	[56]
Wnt	[57,58]

**Table 2 ijms-22-04415-t002:** Biodiverse processes and HS binding protein pathways, modified from.

**A. Go Biological Process Terms Enriched in the Heparin/HS Interactome.**			
**Term**	**Name**	**Count ***	**% ***
GO: 0009611	Response to wounding	120	27.8
GO: 0042330	Taxis	55	12.8
GO: 0006935	Chemotaxis	55	12.8
GO: 0006954	Inflammatory response	73	16.9
GO: 0006952	Defence response	91	21.1
GO: 0007626	Locomotory behavior	62	14.4
GO: 0006955	Immune response	91	21.1
GO: 0042060	Wound healing	51	11.8
GO: 0016477	Cell migration	57	13.2
GO: 0007610	Behavior	71	16.5
GO: 0051674	Localisation of the cell	58	13.5
GO: 0048870	Cell motility	58	13.5
GO: 0042127	Regulation of cell proliferation	90	20.9
GO: 0006928	Cell motion	70	16.2
GO: 0032101	Regulation of response to external stimulus	43	10.0
GO: 0001568	Blood vessel development	51	11.8
GO: 0001944	Vascular development	51	11.8
GO: 0051605	Protein maturation by peptide bond cleavage	33	7.7
GO: 0007267	Cell–cell signaling	76	17.6
GO: 0016485	Protein processing	36	8.4
**B. KEGG Pathways Enriched in the Heparin/HS Interactome.**			
**Term**	**Name**	**Count ***	**% ***
hsa04610	Complement and coagulation cascades	42	9.7
hsa04060	Cytokine-cytokine receptor interaction	63	14.6
hsa04512	ECM-receptor interaction	35	8.1
hsa04510	Focal adhesion	43	10.0
hsa05200	Pathways in cancer	52	12.1
hsa05218	Melanoma	22	5.1
hsa04062	Chemokine signaling pathway	34	7.9
hsa05020	Prion diseases	15	3.5
hsa04810	Regulation of actin cytoskeleton	33	7.7
hsa04350	TGF-β signalling pathway	18	4.2
hsa04672	Intestinal immune network for IgA production	13	3.0
hsa05322	Systemic lupus erythematosus	18	4.2
hsa04010	MAPK signalling pathway	30	7.0
hsa04640	Hematopoietic cell lineage	14	3.2
hsa04621	NOD-like receptor signaling pathway	11	2.6
hsa05219	Bladder cancer	9	2.1
hsa05310	Asthma	7	1.6
hsa05222	Small cell lung cancer	12	2.8

* Count = number of HS binding proteins, % = percentage of the identified proteins.

**Table 3 ijms-22-04415-t003:** Alternative names for FGFRs reflects their diverse areas of interest.

HUGO/MGI Symbol	Receptor Name	Alternative Symbol	Explanation of Symbol
*FGFR1/Fgfr1*	Fgf receptor 1	*Flg* *Flt2* *Cek* *KAL2* *K-sam*	Fms-like gene, ambigiously named since the *FLG* gene encodes filaggrin a skin protein Fms-like tyrosine kinase 2, a gene on chromosome 8p12 encoding FGFR1 Chicken embryo kinase 1, chick cochlea eph class receptor tyrosine kinase [193] Kallman syndrome 2, mutated *FGFR*-1 found in Kallman syndrome 2 [194] KATO III cell-derived stomach cancer amplified gene, sharing homology with *FGFR-1* [195]
*FGFR2/Fgfr2*	Fgf receptor 2	*Bek* *Cek3* *Kgfr*	Bacterial kinase, alias for *FGFR2* Chicken embryo kinase 3, chick fgfr2 Keratinocyte growth factor receptor
*FGFR3/Fgfr3*	Fgf receptor 3	*Cek2*	Chicken embryo kinase 2, chick fgfr3
*FGFR4/Fgfr4*	Fgf receptor 4	*Tkf*	Tyrosine kinase related to FGFR4

**Table 4 ijms-22-04415-t004:** Nuclear HS and HSPGs identified in a number of tumour cell types and in normal cells.

Tumour Type	HS or Proteoglycan Identified	Reference(s)
Bladder carcinoma	HS	[234]
Breast carcinoma	SDC-1	[235]
Glioma	HS, GPC-1	[6]
Chondrosarcoma	SDC-2	[144]
Hepatocyte carcinoma	HS	[236,237]
Lung cancer, Adenocarcinoma	HS, SDC-1	[235]
Melanoma	HS	[238]
Mesothelioma	SDC-1	[204,235]
Monocytic leukemia	HS	[239]
Myeloma	SDC-1	[2,115]
Neuroblastoma	SDC-1	[235]
Nuclear HS and HSPGs identified in normal mammalian cell nuclei		
Cell type	HS or Proteoglycan	Reference
Astrocytes	HS, GPC-2, SDC-2, SDC-3	[240]
Neurons	HS, GPC-1, SDC-2, SDC-3	[6]
Corneal fibroblasts Corneal endothelial cells	HS, HSPG	[114,241,242,243]
Esophageal keratinocytes	HS	[244]
Intervertebral disc cell	Perlecan	[1]

## Data Availability

All data is available in cited references.

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
