# Peer review of "What Are the Potential Roles of Nuclear Perlecan and Other Heparan Sulphate Proteoglycans in the Normal and Malignant Phenotype"

_ijms, 2021, doi:10.3390/ijms22094415_

Round 1

Reviewer 1 Report

The manuscript by Hayes and Melrose is a well written review about the potential role of perlecan expression in the nucleus cells of intervertebral discs, considering not only normal but also tumor cell types.

The Authors did a great literature review, with about 270 citations.

I have no important concerns about the work, but only some minor issues that still remain to be addressed.

  1. Please check and correct some minor spelling mistakes (i.e., affects should be effects in line 141)
  2. In the legend of figure 1, the literature citation n. 267 is included. I guess it should be numbered as citation 34 instead.
  3. Figure 3 is a bit confusing. I think it should be ameliorated to become more readable.

Author Response

Point by Point Responses to Reviewer 1

  • All incorrect “affect” entries have been corrected to effect

  • The reference number has been corrected in the legend to Figure 1

  • Figure 3 has been modified to improve its comprehensibility

Reviewer 2 Report

The authors are expert in effects of perlecan and heparan sulfate, and bring this expertise to this review. The manuscript will benefit by a careful edit, since some of the sentences are just too long to make good sense. There are some irritating grammatical errors, and topic sentences should be placed more deliberately to help the reader. The confocal images of Figure 1 are confusing, since it is not clear what has been modeled. It would be useful to include the original z-stack images, not modeled, and to immunostain the plasma membrane and nuclear membranes, so that the relationship between perlecan and the nucleus is clear. There does not appear to be merged staining of perlecan with DAPI. It would be informative to make clear distinctions between other proteoglycans with predominant heparan sulfate attachments (e.g. syndecan) and perlecan and other sulfated GAGs (e.g. chondroitin sulfates) with heparan sulfate.   Focus on the relationship with malignant phenotype could be improved with a summary of specific attributes of perlecan and of heparan sulfate in relation to recognized hallmarks of cancer, such as those of Weinberg and Hanahan. More clarification about how GAGs stabilize chromatin and precise analysis of how the mechanical properties of the GAGs might contribute to regulation of transcription will help the reader understand the importance of the GAGs.

Author Response

Reviewer 2 Point by Point Responses

Reviewer Comment

The confocal images of Figure 1 are confusing since it is not clear what has been modeled.

Author Response

Additional z-stack confocal images have been added to this figure showing foci of perlecan immunolocalisation within the nucleus, at its periphery and in vesicular forms in the cytoplasm and exported out into the chondron.

Reviewer Comment

It would have been useful to include the original z-stack images, not modeled and to immunostain the plasma and nuclear membrane.

Author Response

Z-stack images have been added.

Reviewer Comment

There does not appear to be merged staining of DAPI and perlecan

Author Response

Perlecan and DAPI staining do not co-localise since perlecan appears to interact with specific structures in the nucleus and we have suggested some possibilities what these structures might be based on known knowledge on HSPGs.

Reviewer Comment

Focus on the relationship with malignant phenotype could be improved with a summary of specific attributes of HS and perlecan in relation to established hallmarks of cancer such as those of Weinberg and Hanahan.

Author Response

We have included a segment on how perlecan is involved in the malignant phenotype as proposed by the hallmark features identified by Weinberg and Hanahan.

Reviewer Comment

More clarification about how GAGs stabilize chromatin and precise analysis of how the mechanical properties of the GAGs might contribute to regulation of transcription will help the reader understand the importance of the GAGs.

Author Response

Comments on GAG-Histone interactions have been added.

Round 2

Reviewer 2 Report

The authors have responded to previous comments and included reference to the hallmarks of cancer and some z-stacks. I would still like to see better demonstration that perlecan is in the nucleus. (The images from Hayes et al 2016 are more convincing, but can not tell thickness of the sections.) There remain some run on sentences and errors in tense and questions about effect vs. affect. The Fig. 1 legend for (b) indicates presence between disc cells, but only one cell is clearly shown. Line 476 is not clear (paracrine autocrine) and L517 and 525 are repetitive. Organization of the manuscript could be tightened to reduce redundancy and emphasize key points.

Author Response

(The authors gave the same response as above.)

Author Response

(The authors gave the same response as above.)

Author Response

(The authors gave the same response as above.)

Author Response

(The authors gave the same response as above.)

Author Response

(The authors gave the same response as above.)

Author Response

Comments on GAG-Histone interactions have been added.

Modified regions are indicated on the revised manuscript in tracked changes mode or are labeled with a new comment label where large regions of the manuscript have been modified.